# Socioeconomic Vulnerability Index and Obesity among Korean Adults

**DOI:** 10.3390/ijerph182413370

**Published:** 2021-12-19

**Authors:** Eunok Park, Young Ko

**Affiliations:** 1College of Nursing, Jeju National University, Jeju 63243, Korea; eopark@jejunu.ac.kr; 2College of Nursing, Gachon University, Incheon 21936, Korea

**Keywords:** socioeconomic status, index, vulnerability, obesity, adults

## Abstract

Examining the socioeconomic vulnerability–obesity relationship is a different approach than comparing obesity rates according to the socioeconomic level. This study explored the socioeconomic vulnerability–obesity relationship among Korean adults. This secondary analysis used data from the Korea National Health and Nutrition Examination Survey, which were collected nationwide from participants aged 30–64 years. Seven socioeconomic indicators (education level, residential area, personal income level, household income level, food insecurity, house ownership, and national basic livelihood security beneficiary status) were used to create the socioeconomic vulnerability index. The prevalence of obesity was higher in the lowest socioeconomic vulnerability index quartile than in the highest socioeconomic vulnerability index quartile (odds ratio = 1.31; 95% confidence interval = 1.13–1.52) after adjusting for gender. When developing future interventions for the prevention and management of obesity, health care providers and researchers need to consider the differences in socioeconomic vulnerability index in adults.

## 1. Introduction

Obesity is associated with cardio- and cerebrovascular diseases, such as dyslipidemia, high blood pressure, and stroke, as well as various health problems, such as arthritis, depression, sleep apnea, and even cancer [1]. In 2015, the number of deaths due to obesity is about 4 million, which is 2.5 times higher than that of non-obese people, and more than two-thirds of the deaths due to obesity are related to cardiovascular disease [2]. The obese population is steadily increasing worldwide. The prevalence of obesity in the adults over 18 years old more than doubled between 1980 and 2014 [3]. In 2016, the adult overweight and obese populations were 39% and 13%, respectively, totaling to 1.0 billion people [4]. Considering the trend and impact of obesity rates, active efforts to prevent and manage obesity are needed. Therefore, the first step in obesity management is to accurately grasp the level of obesity in the population and to understand its epidemiological characteristics.

Socioeconomic vulnerability, defined by socioeconomic status, negatively affects health behavior, lifestyle, and physical and mental health [5]. However, the patterns of obesity within a population group based on the socioeconomic status differ from country to country [6]. In high-income countries, such as the United States, the obesity rate is high in the vulnerable population, whereas in low-income countries, the obesity rate is higher in people with a high socioeconomic level [2]. The income level of personal [7] or household [7,8,9,10] is associated with obesity. A comparison of poor lifestyle habits, such as smoking, drinking problems, and a lack of physical activity, among people with different socioeconomic positions revealed inequality by income or education level using data from the 2013–2015 Korean National Health and Nutrition Examination Survey (KNHANES) in South Korea [11]. A study that analyzed data from the first (1998), fourth (2007–2009), and sixth (2013–2015) KNHANES in South Korea reported that the obesity rate is high in the low-household income group [9]. A meta-analysis of papers on the socioeconomic level–obesity relationship in the United States, United Kingdom, and Canada reported that low-income levels are associated with subsequent obesity, and conversely, obesity is consistently associated with subsequent low-income levels [7]. However, 15-year-old study participants in Nepal from higher-income households had a higher risk of being overweight and/or obese [8]. An individual’s education level also affects their obesity [12,13]. A higher education level reduced obesity [12], and the obesity rate of the low-educated group was higher [13]. However, the effect of socioeconomic status on obesity varies according to the residential area and gender [8,14].

In general, an index combines a series of observable contributing variables into one variable [15]. Since socioeconomic vulnerability is a multidimensional phenomenon, the index generally consists of several subcomponents that aggregate the contributing variables. The socioeconomic vulnerability index (SeVI) is created by aggregating and indexing various socioeconomic indicators that contribute to socioeconomic vulnerability. Previous studies have compared obesity rates according to the socioeconomic level [9,10], but studies on the socioeconomic vulnerability–obesity relationship are rare. Comparing obesity rates using SeVI is a different approach. SeVI can be used to understand the effect of comprehensive socioeconomic vulnerability on obesity. In addition, SeVI can be used as baseline data for developing an obesity intervention program for the underprivileged and for monitoring the effect of the program to promote health equity [15]. Therefore, this study identified the social inequality in obesity due to the socioeconomic vulnerability of adults aged 30–64 years. The aim was to (i) identify the obesity rate among Korean adults; (ii) compare the obesity rate according to demographic characteristics and the seven indicators of socioeconomic status; and (iii) understand the relationship between SeVI and obesity among Korean adults.

## 2. Materials and Methods

### 2.1. Study Design and Participants

This study was a secondary analysis of data from the 2018–2019 KNHANES. Participants were sampled using a two-stage stratified cluster sampling design [16,17]. In all, 7992 people participated in the survey in 2018 and 8110 people in 2019 [16,17], and this study targeted participants aged 30–64 years from the 2018–2019 KNHANES data. Most Koreans in their 20s live with financial support from their parents while attending college or preparing for a job. Therefore, 20s are not the age at which one is expected to lead an independent life with one’s own job. People aged 65 and over are expected to retire from work, representing a population group with characteristics different from those of pre-retirement adults in terms of personal or household income. For these reasons, the subjects of this study were selected as 30–65 years old. Cases of missing values in the variables included in this study, such as the body mass index (BMI) and SeVI were excluded. Therefore, data of 7649 participants were included in the final analysis.

### 2.2. Measure

#### 2.2.1. Obesity

Obesity was assessed using BMI. Trained researchers directly measured height and weight as per guidelines and recorded the measurements to one decimal place [11]. The BMI (kg/m^2^) was calculated using height and weight data and obesity was defined as BMI 25 kg/m^2^ or higher, which is the criteria of Korean adults’ obesity [18,19].

#### 2.2.2. SeVI and Socioeconomic Status

To construct SeVI, the variables used in SeVI in previous studies were reviewed [20]. To evaluate the socioeconomic vulnerability by education level, previous researchers constructed socioeconomic vulnerability index with seven indicators: education level, personal income level, household income level, social activity participation, economic satisfaction, private insurance coverage, and residential area [20]. Each item was assigned a score between 0 and 1, and SeVI was calculated using the sum of these items [20]. Therefore, we reviewed the data survey items collected in the KNHANES, and SeVI was constructed using seven indicators of socioeconomic status: education level, residential area, homeownership, personal income level, household income level, food insecurity, and national basic livelihood security beneficiary status. On the basis of the scoring method of SeVI in previous studies [18], education levels were classified into middle school graduates or lower (1 point), high school (0.5 points), and college or higher (0 points). Residential areas were divided into Eup-Myeon (rural areas) (1 point) and Dong (urban areas) (0 points). For homeownership, 1 point was given for not owning a house and 0 points for owning more than one house. For scoring national basic livelihood security beneficiary status, 1 point was given if the participant was a beneficiary and 0 points if not. Personal and household income levels were divided into lower (1 point), middle–low and middle (0.5 points), and middle–high and higher (0 points), using the quintile. Food insecurity was defined that a lack of consistent access to enough food for an active and healthy life, and it referred to a lack of available financial resources for food at the household level [21]. Food insecurity was assessed by the question “What best represents your family’s food situation in the past year?”; “We were able to eat as much food as we wanted” received 0 points, while “All my family was able to eat enough food, but we couldn’t eat different kinds of food”, “It was economically difficult, so sometimes we were short of food”, or “It was economically difficult, so we often had insufficient food” received 1 point. SeVI was constructed as the sum of all seven items. The higher the score, the higher the socioeconomic vulnerability. As a result of the reliability test of SeVI, Cronbach’s alpha was 0.62. On the basis of the SeVI quartile score, those with a score higher than or equal to the score including the top 25% formed the upper group (SeVI ≤ 0.5), those with a score less than the lower 25% formed the lower group (SeVI ≥ 2.5), and the remaining were classified into the middle group (SeVI = 1–2).

#### 2.2.3. Demographic Characteristics

Demographic characteristics included gender, age, and marital status. Regarding age, the participants were divided into two groups: 30–49 years old and 50–64 years old. The marital status was never married, married, or divorced/separated/widowed.

### 2.3. Data Collection

The 2018–2019 KNHANES was approved by the Research Ethics Review Committee (RERC) of the Centers for Disease Control and Prevention (2018 KNHANES, IRB No. 2018-01-03-P-A; 2019 KNHANES, IRB No. 2018-01-03-C-A) [17,18]. A health survey was conducted through interviews and self-written surveys, a nutrition survey was conducted by the interview survey method, and an examination survey was conducted by direct measurement, observation, and specimen analysis. In compliance with the Privacy Act and the Statistics Act, only non-identifying data were obtained so that participants could not be identified [16,17]. The KNHANES data used are public. We used the raw data released for academic purposes from the website (https://knhanes.kdca.go.kr, accessed on 10 September 2020) [22].

### 2.4. Statistical Analysis

The data were analyzed using SAS 9.4 (SAS, Cary, NC, USA). In all statistical processing, the composite sample weight was applied to reflect the complex sample design when sampling. Since data from 2018 and 2019 were integrated, the combined weight was calculated and used by multiplying the weight, considering the strata, cluster, and survey area, by the integration ratio [17]. The significance level was set to 0.05. The Rao–Scott chi-square test was performed to compare obesity rates according to socioeconomic status. Logistic regression analysis was performed to understand the relationship between SeVI and obesity.

## 3. Results

### 3.1. Demographic Characteristics and Socioeconomic Status

The demographic characteristics and socioeconomic status of the participants are shown in Table 1. About 56% of the participants were 30–49 years old and 43.1% were 50–64 years old. About 50% were men and 49% were women. Regarding marital status, 81.1% were married. University graduates were the maximum (48.0%), while 18.1% had middle school or lower education. About 67% owned more than one house, and 85.6% lived in urban areas, of which 20.0% had low personal income level and 5.3% had low household income level. Regarding food insecurity, 98.8% were able to eat a variety of foods that everyone in their family wanted. Finally, 4.4% were beneficiaries of national basic livelihood security.

The minimum and maximum SeVI values were 0 and 7 points, respectively, with an average of 1.57 points and a standard deviation of 1.23 points. Considering the SeVI distribution, the distribution of each group was confirmed after categorizing the participants into upper (top 25%), middle, and lower (lower 25%) groups. The low-SeVI group comprised 28.7% of the participants, the middle-SeVI group comprised 46.1% of the participants, and the high-SeVI group comprised 25.2% of the participants.

### 3.2. Obesity by Demographic Characteristics and Socioeconomic Status

Table 1 shows the results of comparing obesity rates according to demographic characteristics and socioeconomic status. Obesity rates showed statistically significant differences according to gender, education level, residential area, personal income level, household income level, and SeVI. The obesity rate among men was higher (44.9%) than among women (27.2%). The obesity rate among those who graduated from middle school or lower was significantly higher (39.7%) than among university graduates (34.1%). The obesity rates in urban areas and rural areas were 35.5% and 40.0%, respectively. The obesity rate of the low-income or middle-income of personal income groups was ≥36.0%, which was higher than that of upper income group. The obesity rate was 38.7% in the high-SeVI group, 36.5% in the middle-SeVI group, and 33.5% in the low-SeVI group. Thus, the higher the SeVI, the higher the obesity rate.

### 3.3. SeVI and Obesity

Logistic regression analysis was conducted to identify the relationship between SeVI and obesity. In model 1, only SeVI was treated as an independent variable. In model 2, gender, which showed a significant difference in the comparison of obesity rates according to demographic characteristics, was adjusted and the SeVI–obesity relationship was identified (Table 2). Analysis showed that, compared with the low-SeVI group, the high-SeVI group had a 1.25 times higher risk of obesity (95% confidence interval [CI] = 1.08–1.44) and the middle-SeVI group had a 1.14 times higher risk of obesity (95% CI = 0.99–1.30). After gender adjustment, the high-SeVI group had a 1.31 times higher risk of obesity (95% CI = 1.13–1.52) and the middle-SeVI group had a 1.17 times higher risk of obesity (95% CI = 1.03–1.34). In other words, the risk of obesity increases according to SeVI after adjusting for gender. In addition, men had a 2.20 times higher risk of obesity than women (95% CI = 1.97–2.45).

## 4. Discussion

This study identified the obesity rate according to demographic characteristics and socioeconomic status, in addition to the SeVI–obesity relationship among Korean adults. First, the prevalence of obesity of Korean adults aged 30–64 years was found to be 36.2%. The obesity rate among men is 2.20 times higher than that among women. In 2018, the obesity (standardized) rate among adult men over 30 years old was 44.7%, and that among women was 28.3% [23]. Since the 1998 survey, the obesity rate among men has significantly increased, while the obesity rate among women has remained unchanged [23]. The difference in the obesity rate between men and women increased from 36.6% for men and 28.4% for women in 2008 to 44.7% for men and 28.3% for women in 2018 [23]. Men have a higher prevalence and mortality of cardiovascular diseases related to obesity, such as hypertension, diabetes, and stroke, compared to women [24], so weight management practices for men are important to improve health and reduce the burden of disease.

Second, after adjusting for gender, the high-SeVI group showed a 1.31 times higher risk of obesity than the low-SeVI group. Many Organization for Economic Co-operation and Development (OECD) countries are not only focusing on the rate of increase in obesity and overweightness, but also the inequality in the distribution of obesity among socioeconomic groups [25]. There are various causes of obesity, and in general, daily lifestyle habits, such as high calorie intake and a lack of physical activity, are closely related to obesity [26]. Social and economic factors rather than genetic predisposition contribute more to obesity [9]. Therefore, reducing the socioeconomic inequality in weight-related health problems is becoming a priority in public health. Socioeconomic inequality in the prevalence of obesity can be explained by differences in behaviors that cause obesity, such as eating habits and a sedentary lifestyle [27,28]. However, the effects of socioeconomic class on health behavior may differ by age group in Korea [29]. In this study, we studied adults aged 30–64 years in this study. In the future, it is necessary to conduct research on the socioeconomic vulnerability–obesity relationship in other age groups such as children, adolescents, and older adults.

Third, among the seven indicators included in SeVI, the low education level, rural residence, and low personal income level were associated with obesity. These results are similar to those of previous studies [30,31]. In a previous study conducted in the United States, it was reported that those with a lower education level (below middle school) have a 1.8 times higher risk of obesity than those with university-level or higher education, while those with a lower income level have a 1.5 times higher risk of obesity than those with a high-income level [30]. In a systematic review of education level and obesity, the education level–obesity relationship differed by the socioeconomic level of the country [31]. In developed countries, the lower the level of education, the higher the risk of obesity, whereas in developing countries, the higher the level of education, the higher the risk of obesity [31]. Therefore, when examining the relationship between SeVI and obesity, it is important to consider the socioeconomic context of the country.

We found no significant difference in the obesity rate based on homeownership, food insecurity, household income, and national basic livelihood security beneficiary status. Several researchers also reported that there was no statistical relationship between homeownership and obesity [30]. Although homeownership reflects the socioeconomic level, its relationship with obesity was not significant in this study, probably because housing ownership conditions in urban and rural areas are different and the obesity rate in rural areas is high. In other words, in rural areas, even if the economic level is low, more people may own houses, while in large cities, even if the economy is relatively good, people may not own houses. The proportion of participants with food insecurity and nation basic livelihood security was relatively small, with 1.2% of participants with food insecurity and 4.4% being beneficiaries of nation basic livelihood security. Therefore, the relationship with obesity was not statistically significant, even though the obesity rate in these groups was relatively high at 40.8% and 37.7%, respectively, so it is necessary to focus on their obesity as well.

In this study, we used data collected as representative samples. The BMI was calculated using values measured by standardized guidelines for height and weight through a medical examination to determine obesity. Therefore, this study is significant in that it estimates the obesity rate of adults aged 30–64 years on the basis of relatively accurate data. In addition, SeVI is an aggregate measure of socioeconomic status indicators at the individual level and consists of several socioeconomic indicators. Income and education are the most widely used indicators to measure socioeconomic status at the individual level [7,8,9,10,12,13]. Other socioeconomic indicators, such as household income and assets [32], an individual’s residential neighborhood [28], residential property values [33], and homeownership [34], are also reported to influence obesity among adults in developing countries. Therefore, it is significant in that SeVI was calculated using several indicators and that the obesity rate and the risk of obesity by SeVI groups were identified. This study has some limitations. Since data were collected cross-sectionally, the SeVI–obesity relationship could not be interpreted as a cause-and-effect relationship. We constructed SeVI using the variables at the individual-level from the original KNHANES data. Community-level factors, such as neighborhood environment and accessibility to unhealthy food and physical activity resources [35], may determine socioeconomic vulnerability and influence obesity. We did not consider the influences of community-level variables on obesity in this study. Future studies need to identify the association between community-level SeVI and obesity.

## 5. Conclusions

This study identified the obesity rate and obesity risk according to SeVI using data collected as a representative sample. Even after adjusting for gender, the group with high socioeconomic vulnerability had a significantly higher obesity rate than the group with low socioeconomic vulnerability. The education level, residential area, and personal income level were associated with obesity. These research results show that there is a need to (i) improve health equity in obesity; (ii) use and monitor SeVI in developing interventions for obesity prevention and management or evaluate their effectiveness; and (iii) develop a comprehensive SeVI and identify an association between individual dimensions of SeVI with obesity.

## Figures and Tables

**Table 1 ijerph-18-13370-t001:** Prevalence of obesity by demographic characteristics and socioeconomic status. *N* = 7649.

Variables	Categories	*n*	%	Obesity Weighted %	Chi	*p*-Value
Total		7649	100	36.2		
Age group	30–49 years	4074	56.9	36.1	0.02	0.903
	50–64 years	3575	43.1	36.3		
Gender	Men	3339	50.7	44.9	188.99	<0.001
	Women	4310	49.3	27.2		
Marital Status	Never married	758	11.1	36.0	3.27	0.195
	Married	6200	81.1	36.6		
	Divorced/separated	691	7.7	32.2		
Education	University	3535	48.0	34.1	13.41	<0.001
	High school	2606	33.9	37.2		
	Middle school	1508	18.1	39.7		
Housing	Own	5175	67.4	35.8	0.74	0.389
	Rent	2474	32.6	36.9		
Residential region	Urban	6357	85.6	35.5	5.02	0.025
	Rural	1292	14.4	40.0		
Individual income	Upper	3054	38.8	33.9	10.27	0.006
	Middle	3081	41.2	38.4		
	Lower	1514	20.0	36.0		
Household income	Upper	4281	56.1	35.2	4.83	0.090
	Middle	2925	38.5	37.8		
	Lower	443	5.3	34.8		
Food insecurity	Enough	7548	98.8	36.1	0.83	0.363
	Lack	101	1.2	40.8		
Recipients of national basic livelihood security	No	7287	95.6	36.1	0.29	0.591
	Yes	362	4.4	37.7		
SeVI	≤0.5	2159	28.7	33.5	9.36	0.009
	1~2	3469	46.1	36.5		
	≥2.5	2021	25.2	38.7		
	M ± SD (0–7)	1.57 ± 1.23				

Note. M ± SD, mean ± standard deviation; SeVI, socioeconomic vulnerability index.

**Table 2 ijerph-18-13370-t002:** Odds ratio for socioeconomic vulnerability index on obesity. *N* = 7649.

Variables	Category (Range)	Model 1			Model 2		
		Odds Ratio	95% CI	*p*-Value	Odds Ratio	95% CI	*p*-Value
SeVI Group	SeVI ≤ 0.5	1			1		
	1 ≤ SeVI≤ 2	1.14	0.99~1.30	0.051	1.17	1.03~1.34	0.020
	2.5 ≤ SeVI	1.25	1.08~1.44	0.002	1.31	1.13~1.52	<0.001
Gender	Women				1		
	Men				2.20	1.97~2.45	<0.001
F (*p*)		5.03 (0.007)			73.14 (<0.001)		

Note. SeVI, socioeconomic vulnerability index; CI, confidence interval.

## Data Availability

The data that support the finding of this study was available from the corresponding author, upon reasonable request.

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
