# Peer review of "Socioeconomic Vulnerability Index and Obesity among Korean Adults"

_ijerph, 2021, doi:10.3390/ijerph182413370_

Round 1

Reviewer 1 Report

Overall comment

I think you need to better motivate the use of the socioeconomic vulnerability index (SeVI). You say it's a "different approach" (page 2, line 58), but you don't state clearly why it's useful to use an index of various factors rather than all of them separately. For a policymaker or a public health program designer, at the end of the day, they will care about which components of the index are the main drivers of the SeVI-obesity relationship. If it is mostly low education, for example, the policy implication would be that improving education among low-educated populations may lead to beneficial weight outcomes. The summation of multiple SES indicators allow you to get a wide continuum of disadvantage, so it allows you to estimate how much obesity risk increases with greater SES disadvantage and shows that disadvantages can have an additive effect on obesity risk. That is useful, but it is most useful when you can show the component or components that mostly explain that relationship. That way, it is clear what disadvantages need to be addressed to reduce obesity risk. 

Specific comments

  • I recommend heavily revising the last sentence in the Abstract as it is not supported by your results: “An obesity prevention and management program for the socioeconomically vulnerable population must be developed and implemented to reduce health inequality in obesity.” Your paper estimates the association between a SES index and obesity prevalence, it does not show evidence that an obesity-related program can reduce differences in obesity rates by SES.
  • Introduction, line 25-26: 4 million deaths over what year or period of time?
  • Introduction, line 39: Whose income level? I think you meant to say that a person's or household's income level has been shown in the literature to be associated with obesity. 
  • Introduction, line 51: Whose education level? I think you meant to say that a person's education level been shown in the literature to be associated with obesity.
  • Page 2, line 70: Why does the analysis focus on adults 30+ and not all adults 20+? Also is there are reason you want to not include 65+? I assume it's because height drops off among the elderly? Please also mention here whether the height/weight data are measured by a health professional or self-reported. Are 2018-2020 data available? The results would be most useful if they cover the most recent time period possible.  
  • Page 2, line 78: Isn’t the obesity cutoff among Asian persons a BMI of 27.5+ (https://doi.org/10.2337/dc14-2391)? Generally, a body mass index (BMI) 25+ is considered overweight, and 30+ is obese. However the cutoffs are different for Asian populations, so I recommend that you redo the analysis with an obese indicator using the most relevant BMI cutoffs for Korean individuals.
  • The table 2 title is not correct, it must be from a different paper. Also is the sample size correct in table 2? Shouldn’t it be lower as in table 1 (N = 7,638)
  • For table 2, I recommend adding another model (model 3) in which you control for all demographic characteristics (age group, gender, and marital status). Also are there any other important demographic info in your survey data such as ethnic group, immigrant status, and presence of children? If so I would include this info in your Table 1 and Table 2 (model 3).
  • Discussion, line 186: You found that 36.5% of your sample was obese. Where does "4 of 10 of the participants were obese" come from? Seems like you rounded up and repeated the info in the sentence?
  •  Discussion, line 192: I recommend toning down the language. Instead of saying "it is necessary to manage obesity" you might say weight management practices are important to improve health and reduce disease burden (along with appropriate citations)
  • Discussion, line 223: What do you mean it is necessary to analyze...according to gender? If you meant it's important to estimate gender-specific estimates of the relationship, you could do that in your data (e.g. by running the logit regression among men and women separately)

Author Response

Dear Reviewer

We wish to thank you for your thoughtful comments and valuable feedback on the manuscript entitled “Socioeconomic Vulnerability Index and Obesity among Korean Adults.” We would like to resubmit the revised manuscript for publication in the International Journal of Environmental Research and Public Health.

We have tried to revise the manuscript according to your suggestions and rewrote or rephrased sections to improve clarity. For your convenience, we have used red font for the revisions. Please find the following revisions according to reviewer’s comments.

Further, I believe that this revised paper will be of interest to the readership of the International Journal of Environmental Research and Public Health. Thank you for your consideration. I look forward to hearing from you.

Reviewer 2 Report

The absence of limitations and future research ambitions in this area is considered a shortcoming.

Overall, I evaluate the study very positively and the above-mentioned shortcomings are in principle only of a formal nature.

Author Response

(The authors gave the same response as above.)

Reviewer 3 Report

This paper describes a new composite indicator, socio-economic vulnerability and its relationship with obesity. It takes an interesting approach to understanding socio-economic status at the individual level, however the paper has very limited conceptual set-up/framework for why this type of indicator is needed. Particularly when compared to simple income thresholds that are used for policy and programming, particularly in the U.S. context. More details need to be provided about how this measure can be used to inform obesity prevention programs. 

Author Response

(The authors gave the same response as above.)

Round 2

Reviewer 1 Report

Thank you for addressing each of my comments. 

Author Response

Thank you so much.

Reviewer 3 Report

A thorough justification for why this composite indicator is important when compared to simple income thresholds is not provided. Additionally, new sentences added in red (p. 7, line 261 to 267) are problematic. For instance in the first sentence (line 261) you state that because the dataset is missing many indicators that the paper is indeed not(?) capturing the main indicators for socioeconomic vulnerability. This sentence seems to say that the KNHANES is not the appropriate dataset for answering this research question. Second, line 265, states that were not able to determine the dimensions of SVI contribute to obesity. This statement contradicts the whole discussion section.

Author Response

Thank you so much for your comments. We have revised the limitation as comments. We have deleted some sentences that may cause misunderstandings and have revised the limitation clearly.